# A Novel Physically Based Distributed Model for Irrigation Districts' Water Movement

Boyu Mi [1,2], Haorui Chen [1,2,*], Shaoli Wang [1,2], Yinlong Jin [3], Jiangdong Jia [4], Xiaomin Chang [1,2], Xiaojun Fu [5], Ronghua Chai [5] and Meiling Wei [5]

1   State Key Laboratory of Simulation and Regulation of Water Cycle in River Basin, China Institute of Water Resources and Hydropower Research, Beijing 100038, China; miboyu@yeah.net (B.M.); shaoliw@iwhr.com (S.W.); xmchang2013@163.com (X.C.)
2   National Center for Efficient Irrigation Engineering and Technology Research-Beijing, Beijing 100038, China
3   School of Water Resources and Hydropower Research, Wuhan University, Wuhan 430072, China; wrhjinyl@whu.edu.cn
4   College of Water Resources and Architectural Engineering, Northwest A&F University, Yangling 712100, China; jiangdong0501@nwafu.edu.cn
5   Shahaoqu Station, Jiefangzha Irrigation Area Management Bureau, Hetao Irrigation District, Inner Mongolia 015400, China; fuxiaojun0303@163.com (X.F.); chaironghua0303@163.com (R.C.); shqsyz2017@163.com (M.W.)
*   Correspondence: chenhr@iwhr.com

**Abstract:** The water movement research in irrigation districts is important for food production. Many hydrological models have been proposed to simulate the water movement on the regional scale, yet few of them have comprehensively considered processes in the irrigation districts. A novel physically based distributed model, the Irrigation Districts Model (IDM), was constructed in this study to address this problem. The model combined the 1D canal and ditch flow, the 1D soil water movement, the 2D groundwater movement, and the water interactions among these processes. It was calibrated and verified with two-year experimental data from Shahaoqu Sub-Irrigation Area in Hetao Irrigation District. The overall water balance error is 2.9% and 1.6% for the two years, respectively. The Nash–Sutcliffe efficiency coefficient (NSE) of water table depth and soil water content is 0.72 and 0.64 in the calibration year and 0.68 and 0.64 in the verification year. The results show good correspondence between the simulation and observation. It is practicable to apply the model in water movement research of irrigation districts.

**Keywords:** hydrological model; irrigation districts; irrigation and drainage; soil water; groundwater; unstructured grid; Shahaoqu Sub-Irrigation Area

## 1. Introduction

Irrigation districts play an important role in China's agricultural production, food safety, and social economy [1]. A better understanding of water movement in irrigation districts, especially in arid and semi-arid areas, is a prerequisite for irrigation and drainage management with high efficiency of water usage. Hydrological modeling, as a supplement to the theoretical and experimental study at the beginning, is undergoing rapid development due to the accelerated progress of computer science and numerical techniques [2]. Tremendous efforts have been made for the development and application of hydrological models, of which many were adopted in the study of water movement in irrigation districts directly or in a combinative or improved way. Irrigation districts are quite different from usual regions considered in generalized hydrological models. There are more artificial regulations in irrigation districts to control the water flow, which complicates the water distribution and movement compared to natural regions. The water distribution through irrigation canals is controlled by the gate system, which is hard to describe in a hydrological

model focusing on natural flow, not to mention the differential irrigation as a result of divergent cropping land use. The water in arid and semi-arid irrigation districts mainly comes from irrigation, which increases the impact of irrigating process on the regional water cycle. Crops grow in the unsaturated soil surface layer, and the water movement is arduous to simulate with the 3D Richards equation at the regional scale [3], which brings in challenges. The water exchange between surface water, soil water, and groundwater, together with complex boundary conditions such as meteorology information, water absorption by crop roots, and irrigation and drainage management, makes the water movement in irrigation districts an intricate system.

Researchers have tried to characterize such a system in numerous ways. SWAT is a widely used semi-distributed hydrological model based on the geographic information system (GIS) [4], and modification is required when applied to irrigation districts. Zheng et al. [5] modified the extraction of drainage ditches, distributed subbasins, and hydrological response units, as well as the calculation method of the crop's actual ET, in a SWAT model, and used it to simulate and analyze the water balance in an irrigation district. Xie et al. [6] incorporated processes for irrigation and drainage into SWAT. However, this model is hydrologic-process-based and driven by water balance, and the soil water and groundwater movement are oversimplified. The same applies to VIC, another semi-distributed hydrological model, which balances both the water and surface energy within the grid cell [7]. CLM is widely used in land surface hydrologic process simulation [8]. By coupling with other models, it was introduced to irrigation studies [9]. This model describes surface water flow and soil water movement in detail while lacking consideration of groundwater flow. The dynamic TOPMODEL is physically based and can be used to simulate water flux through a watershed, including the interactions between groundwater and surface water [10]. However, the irrigation and drainage processes are not well considered. SWMM is mainly used for the simulation of urban runoff quantity and quality [11]. It considers the surface drainage, including the influence of a water pond, water diversion, gate, and pump, and has been successfully used in irrigation and drainage research [12]. However, the soil water and groundwater movements are not contained, which are important for irrigation districts. SIMODIS is a spatially distributed agro-hydrological model [13], which can be used to assess the water condition in fields and aid the decision-making system for irrigation management [14]. However, this model focuses on the soil water part, and the surface flow and groundwater movements are not considered. MODFLOW [15] and FEFLOW [16], prominent in groundwater simulation, are extensively applied to groundwater movement research in irrigation districts [17,18]. However, the surface flow is usually generalized into boundary conditions. Hydrus-2D/3D can simulate subsurface flow in both saturated and unsaturated soil [19], but the surface flow is not contained, and it is strenuous to apply a 3D soil water model at the regional scale. Zhu et al. [20] coupled the 1D unsaturated flow with the 3D saturated flow to simplify the soil water movement and made the subsurface flow simulation acceptable on a large scale. However, the surface flow was not considered. Most of the regional hydrological models are designed for general purposes, and can only deal with part of the processes in irrigation districts. Researchers tried coupling different models to cover more processes. Kim et al. [21] integrated SWAT with MODFLOW to generate a detailed representation of groundwater recharge. Aliyari et al. [22] extended the SWAT-MODFLOW model to large agro-urban river basins in semi-arid regions. However, the hydraulic process in crop root zoon is still not clear. Some field-scale models, such as HYDRUS-1D [23], SWAP [24], and SHAW [25], have immense applications in irrigation fields [26–28]. They describe 1D soil water movement in detail and can be extended to a regional scale by coupling with GIS [29] and models dealing with the surface flow and/or groundwater movement [30–33]. However, these models cannot describe the irrigation process well. Even some comprehensive models fail to take into account the water diversion process in irrigation districts, which is highly affected by manual regulations. MIKE SHE fully considers surface flow, soil water, and groundwater movement [34]. HydroGeoSphere is also a fully integrated, physically

based hydrological model [35]. However, they are not designed for irrigation districts, and the treatments of the irrigation process and crop absorption are not satisfactory. WaSiM-ETH uses the Richards equation for describing soil water movement and a 2D groundwater flow model for layered aquifers in its second version [36]. It also considers irrigation with surface water as well as pumped groundwater, artificial drainage, and crop ET. However, the water diversion process is not described. SIMGRO covers the whole processes in a regional system, including plant–atmosphere interactions, surface water, soil water, and groundwater [37], which is suitable for studies in irrigation districts. However, there is no irrigation process through canals, which means the water diversion process and the corresponding canal seepage cannot be characterized.

There have been many models describing water movement at the regional scale, with both surface flow and subsurface flow. However, it is rare to see a model specialized for irrigation districts, with comprehensive consideration of the irrigation process, soil water movement, water absorption by crop roots, groundwater movement, drainage process, and water interactions among these processes. The former research mainly focuses on surface flow, groundwater flow, or both, but not soil water movement, which are all important processes in irrigation districts, especially the soil water movement that influences crop growth. The former consideration for surface flow is usually driven by a Digital Elevation Model (DEM), which is acceptable for a natural river system, but not for the canals and ditches in irrigation districts that are highly regulated. In this paper, we constructed a novel distributed model that is based on hydraulics and has a full consideration of all the important processes in irrigation districts to make up for the deficiencies of previous studies. The model is driven by the topology of canals, ditches, and fields, which can satisfyingly describe the surface flow in irrigation districts.

## 2. Methodology

The most significant characteristic of irrigation districts, compared to natural regions, would be artificial interference, including irrigation and drainage management, and agricultural land use. The focal point of water movement research is on the crop root layer, which is strongly affected by the surface water and the groundwater. The crop root zone is usually unsaturated, which can be described by the 3D Richards equation and the soil-water characteristic curve, as did in HYDRUS-2D/3D and HydroGeoSphere. However, the computational consumption is unbearable when applied at the regional scale for at least one period of crop growth. Considering that the water moves primarily in the vertical direction before reaching the groundwater level, a compromise could be made to simplify the unsaturated zone above groundwater level to parallel vertical columns and then couple them with the groundwater model, as did in SIMGRO and Zhu's model [20]. As for the groundwater movement, the Dupuit assumption is adopted in this study, which has been testified by Yue et al. [38]. It assumes that the groundwater level is relatively flat and the vertical movement can be omitted, leading to a simplified 2D horizontal flow. The surface flow mainly occurs in irrigation canals and drainage ditches, which is typical of open-channel flow, and the mechanism can be described by the 1D diffusive wave equation. The interactions among different parts are illustrated in Figure 1.

### 2.1. Soil Water Movement

The unsaturated soil zone is divided into independent soil columns, without horizontal water exchange. The soil water movement in each column is governed by the 1D Richards equation in the vertical direction, as shown in Equation (1):

$$\frac{\partial \theta}{\partial t} = \frac{\partial}{\partial z}\left(K\frac{\partial h}{\partial z} + K\right) + S_{soil} \tag{1}$$

where $\theta$ is the water content [-], $t$ is the time [T], $z$ is the vertical axis [L], $h$ is the pressure head [L], $K$ is the hydraulic conductivity [L·T$^{-1}$], and $S_{soil}$ is the source in unit depth [T$^{-1}$] (negative for outflow).

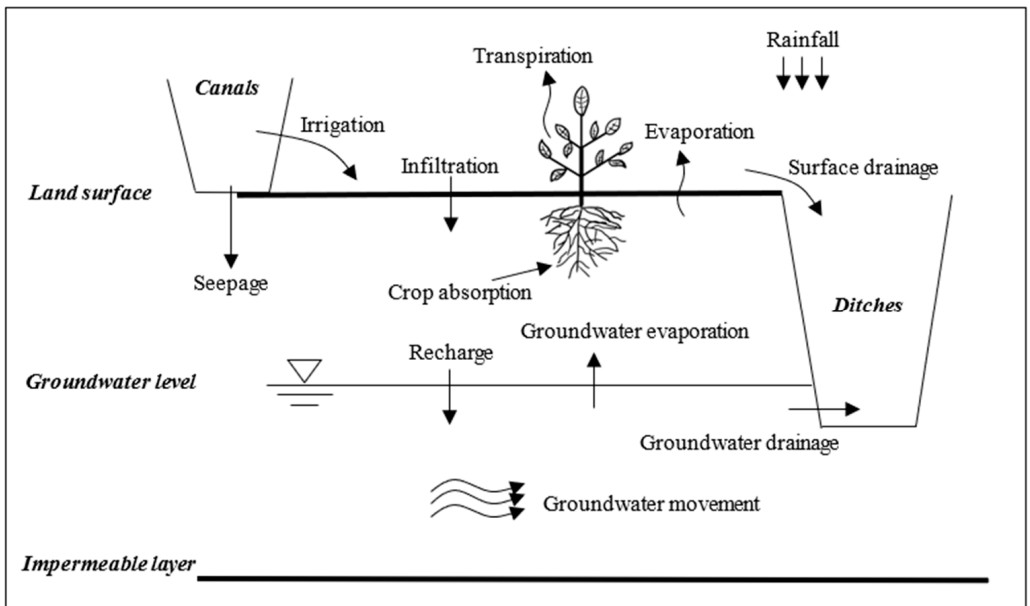

**Figure 1.** Water interactions among surface water and subsurface water in irrigation districts.

The equation is closed by the van Genuchten (VG) model [39], a widely used empirical formula for describing the soil–water characteristic curve, and an iterative method is involved in synchronizing $\theta$, $h$, and $K$ at each time step. The time step is dynamically adjusted according to the iterations at the last time, which is learned from HYDRUS-1D. The spatial discretization is a set of evenly distributed vertical grids. The top grid is located on the land surface, and the bottom grid is located below the groundwater level.

The lower boundary is the groundwater level, which comes from the groundwater module and will be transformed to the pressure head at grids below it. Thus, the bottom grid should always locate below the dynamic groundwater level. The upper boundary is coupled with the surface ponding balance. Assuming there is a layer of water on the land surface, the water balance equation will be

$$\frac{\partial H_{surf}}{\partial t} = Prec + Irri - Evap - Drain - intch + S_{surf} \tag{2}$$

where $H_{surf}$ is the surface ponding depth [L], which can also be interpreted as the pressure head on the top grid, $t$ is the time [T], *Prec* is the precipitation rate [L·T$^{-1}$], *Irri* is the irrigation rate [L·T$^{-1}$], *Evap* is the evaporation rate [L·T$^{-1}$], *Drain* is the surface drainage rate [L·T$^{-1}$], *intch* is the downward interchange rate through the land surface [L·T$^{-1}$] (positive for infiltration), and $S_{surf}$ is the source rate [L·T$^{-1}$] (positive for incoming source). The interchange rate *intch* is exactly the upper boundary flux of soil water, which means Equation (2) can be integrated into the upper boundary of Equation (1) through substituting the upper boundary flux with Equation (3), a variation of Equation (2):

$$intch = PIEDS - \frac{\partial H_{surf}}{\partial t} \tag{3}$$

where *PIEDS* = *Prec* + *Irri* − *Evap* − *Drain* + $S_{surf}$. The ponding depth (or surface pressure head) $H_{surf}$ will be solved with soil pressure heads at other grids simultaneously. If the surface is dry, the ponding depth becomes zero, and it is no longer the surface pressure head, then Equation (3) becomes

$$intch = PIEDS \tag{4}$$

in which *intch* could be positive or negative. A negative *intch* means the land surface evaporation and the evaporation rate will be limited by soil water condition at the surface grid; a positive *intch* indicates infiltration, and if *intch* is greater than the maximum possible infiltration rate, surface ponding begins.

The consideration for root water uptake is much the same as HYDRUS-1D. The actual transpiration (water absorption) rate is distributed along the root zone, as a function of location (considering root density) and real-time pressure head (considering water stress):

$$T_a = \int TD_a(z,h)\mathrm{d}z = \int \alpha(h)TD_P(z)\mathrm{d}z = \int \alpha(h)\beta(z)T_P\mathrm{d}z \tag{5}$$

where $T_a$ is the actual transpiration rate of the root zone [L·T$^{-1}$], $TD_a$ is the actual transpiration rate distribution [T$^{-1}$], which is a function of elevation $z$ [L] and the corresponding pressure head $h$ [L], $TD_p$ is the potential transpiration rate distribution [T$^{-1}$], $T_p$ is the potential transpiration rate of the root zone [L·T$^{-1}$], $\alpha(h)$ is the water stress function [-], and $\beta(z)$ is the distribution function relating to the root density [L$^{-1}$].

The water stress function is given by van Genuchten [40], without consideration for the osmotic head:

$$\alpha(h) = \frac{1}{1 + \left(\frac{h}{h_{50}}\right)^p} \tag{6}$$

in which $p$ is approximately 3, and $h_{50}$ is the pressure head at which the transpiration rate is reduced by 50%.

The distribution function is given by Hoffman et al. [41]:

$$\beta(z) = \begin{cases} 0, & z \geq Z_{top} \\ \frac{5}{3RD}, & Z_{top} - \frac{RD}{5} \leq z < Z_{top} \\ \frac{25}{12RD}\left(1 - \frac{Z_{top}-z}{RD}\right), & Z_{top} - RD \leq z < Z_{top} - \frac{RD}{5} \\ 0, & z < Z_{top} - RD \end{cases} \tag{7}$$

in which $Z_{top}$ is the land surface elevation [L], and $RD$ is the root depth [L].

The potential transpiration rate $T_p$ is calculated through Equation (8):

$$T_P = K_T \cdot ET_P = K_T \cdot K_C \cdot ET_0 \tag{8}$$

where $K_T$ is the transpiration coefficient [-], $ETp$ is the crop potential evapotranspiration rate [L·T$^{-1}$], $K_C$ is the crop coefficient [-], and $ET_0$ is the potential evapotranspiration rate of reference crop [L·T$^{-1}$]. The $ET_0$ is calculated through meteorological information with the Penman–Monteith equation [42]. The $K_C$ is related to the crop and given as a parameter. The $K_T$ is calculated from the Leaf Area Index (LAI) [43]:

$$K_T = 1 - e^{-k \cdot LAI} \tag{9}$$

in which $k$ is the light extinction coefficient.

### 2.2. Groundwater Movement

The groundwater movement is simplified to a 2D horizontal flow. Considering the irregular pattern of land use in irrigation districts, especially the branched distribution of canals and ditches, the spatial discretization for groundwater flow is a set of unstructured grids (or cells) with any shape, while triangles and quadrangles are adequate to cover most cases and approximate regular polygons are strongly suggested to minimize the interpolating deviation. The governing equation of 2D groundwater flow is given by Equation (10):

$$\mu \frac{\partial Z_{grnd}}{\partial t} = \nabla \cdot \left(K_s \nabla Z_{grnd} H_{grnd}\right) + S_{grnd} \tag{10}$$

where $\mu$ is the specific yield or specific storage [-], $Z_{grnd}$ is the groundwater level [L], $t$ is the time [T], $\nabla$ is the gradient operator [L$^{-1}$], $Ks$ is the saturated hydraulic conductivity [L·T$^{-1}$], $H_{grnd}$ is the thickness of the groundwater [L], and $S_{grnd}$ is the source in unit horizontal area [L·T$^{-1}$] (positive for inflow). Considering a cell of any shape in unstructured grids, the governing equation can be rewritten as Equation (11):

$$\mu \frac{\partial Z_{grnd}}{\partial t} = \sum_{j=1}^{n} \left( K_s \frac{\partial Z_{grnd}}{\partial r_j} \frac{A_j}{A_{cell}} \right) + S_{grnd} \tag{11}$$

where $n$ is the number of cell sides [-], $r_j$ is the outer normal direction of the $j$-th side [L], $\frac{\partial Z_{grnd}}{\partial r_j}$ is the outer normal gradient of the groundwater level at the $j$-th side [-], $A_j$ is the vertical cross area at the $j$-th side [L$^2$], and $A_{cell}$ is the horizontal area of the cell [L$^2$]. $\left( K_s \frac{\partial Z_{grnd}}{\partial r_j} A_j \right)$ is the inflowing flux through the $j$-th side. The groundwater level change is a result of the side flow and other sources such as soil water recharge, canal seepage, and drainage.

The normal gradient of groundwater level at the cell side is crucial to the flux calculation. It is a good approximation by using the difference quotient of adjacent cells in orthogonal grids, but an error will be introduced for unstructured grids in this study. Take three cells in Figure 2 as an example. To get the gradient through edge $ab$, which is between cell $i$ and $j$, using the groundwater level at interpolated point $i'$ and $j'$, located on the bisector $cd$ of edge $ab$, will be better than using that of cell centroid $i$ and $j$ directly, which has been confirmed by MODFLOW-USG [44], an unstructured grid version of MODFLOW. The method of finding these interpolated points (or ghost nodes in MODFLOW-USG) is improved in this study. The groundwater level $Z_{grnd}$ at the interpolated point (say $Z_{gi'}$) is a weighted average of $Z_{grnd}$ at cell centroids nearby (say $Z_{gi}$, $Z_{gj}$, and $Z_{gk}$). The weighting factors are given additionally, and the number of weighting factors remains the same for all ghost nodes in MODFLOW-USG, while it is automatically analyzed from grid information in this study. Cell centroids within the local range of the ghost node will be used as interpolating points, and the weighting factors are calculated with an improved Inverse Distance Weighted (IDW) method, a spatial interpolating method that takes into account the azimuth of the interpolating centroid as well as its distance, which is better than the traditional IDW method that only considers the distance.

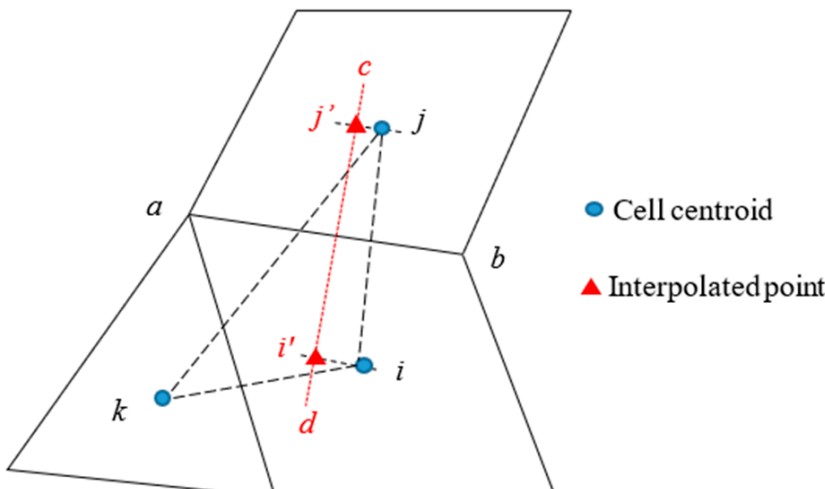

**Figure 2.** Example cells of unstructured grids.

### 2.3. Channel Flow

The water flow in canals and ditches are both open-channel flow, which can be described by the 1D diffusive wave equation, as shown in Equation (12):

$$B_{ch}\frac{\partial Z_{ch}}{\partial t} = \frac{\partial}{\partial x}\left(D_{ch}\frac{\partial Z_{ch}}{\partial x}\right) + q_{ch} \tag{12}$$

where $B_{ch}$ is the water surface width in the channel [L], $Z_{ch}$ is the channel water level [L], $t$ is the time [T], $x$ is the longitudinal direction [L], $q_{ch}$ is the source in unit length [$L^2 \cdot T^{-1}$] (positive for inflow), and $D_{ch}$ is the diffusive coefficient [$L^3 \cdot T^{-1}$], as given in Equation (13):

$$D_{ch} = \frac{A_{ch}R_{ch}^{2/3}}{N_m\sqrt{\left|\frac{\partial Z_{ch}}{\partial x}\right|}} \tag{13}$$

in which $A_{ch}$ is the cross area [$L^2$], $R_{ch}$ is the hydraulic radius [L], and $N_m$ is Manning's roughness coefficient [$L^{-1/3} \cdot T$].

Different levels of channels connect to form a functional network for irrigation or drainage. The canal network and ditch network are usually branch-like, but with opposite flow patterns, one for diverging water by levels to the fields, and one for converging water by levels from the fields. The ditch network is close to the natural river basin, which is considered in most models describing the surface flow. However, the canal network is special because it appears in an artificial way, which requires extra consideration such as the water diverging method.

Since the canal network and ditch network are special cases of the channel network, a generalized channel flow is modeled in this study to cover features in both canal flow and ditch flow. Figure 3 shows a generalized representation of the channel network. The network is composed of connected chains, which will be discretized into segments, each containing a cross section. The diffusive wave equation is applied on these chains, which will then be coupled through the connecting points, either diverging points for canals or converging points for ditches.

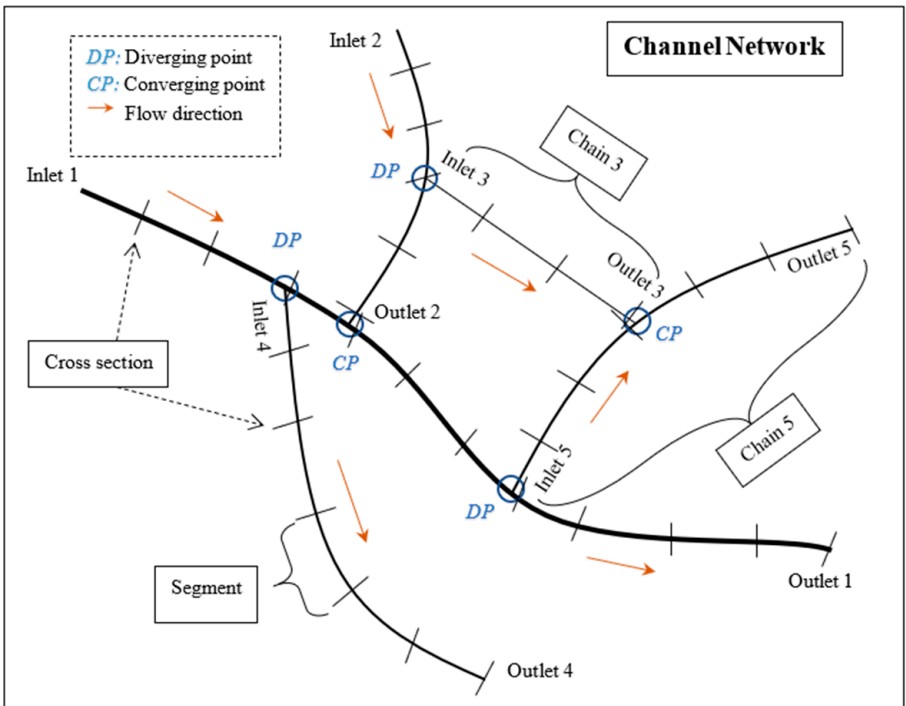

**Figure 3.** A generalized representation of the channel network.

The irrigation process is discontinuous during the annual cycle, which means water flows on canals at certain times. A complete irrigation round begins from the opening of the headgate and ends at the drying up of all canals. Usually, there will be several rounds in a year depending on the soil water condition and irrigation management. The simulation of canal water flow will only be activated during an irrigation round to save computing time. The irrigation water comes from the head inlet(s) of the canal network and flows through all possible canal chains until being delivered to the fields. During the irrigation process, part of the water may evaporate to the air and seepage to the groundwater, and redundant water may be discarded to the drainage ditches. The water transferred from canal segments to their connected fields has another seepage portion through the lower-level canals that are too small to be characterized in the canal network. The evaporation data are from the meteorological information, and the seepage is controlled by the canal water usage efficiency. The groundwater cells are mapped to the land surface, and each of the surface cells represents a generalized field, within which there could be one or more kinds of land use. The irrigation water distribution is controlled by the land use in the field as well as the segment–field connection, which means the water delivery from a canal segment to a field varies with space (canal distribution) and time (only the selected crops will be irrigated in an irrigation round).

On the other hand, the drainage process is continuous and simpler. The ditch segments exchange water with the surface cells and the groundwater cells, depending on the water level difference, and the exchanged water is added to the source term of Equation (12). The Hooghoudt formula [45] is used to calculate the groundwater drainage rate. There is also evaporation on ditches, and the discarded water from canals will flow into the ditches at connecting points.

*2.4. Model Coupling*

The coupling of soil water movement, groundwater movement, and canal and ditch flow is through their interactions shown in Figure 1, which can be described as spatial connections. From the perspective of spatial discretization, the irrigation district is first divided into small polygons horizontally, according to the land use and canal and ditch distribution, as shown in Figure 4. Each polygon represents a cell for groundwater, soil water, and surface field, with a unique ID. A soil water cell and the corresponding surface cell could be subdivided into several units with different proportions if there is more than one kind of land use, which means the soil water unit and the surface unit are bound together and a groundwater cell may connect with one or more soil water units, each representing a soil column for the soil water movement simulation. The soil water units are independent of each other, but the units in a cell will be assembled before interchanging with the corresponding groundwater cell. The groundwater cell provides a groundwater level to the corresponding soil water units and receives their recharge amount in return. The surface unit water balance is integrated into the upper boundary of the soil water unit.

The canal network and ditch network spreading on the land surface have complex connections with surface cells and groundwater cells. The network is composed of chains, which will be divided into segments. A practical way is to cut the line with groundwater cell polygons, which will result in good correspondence between these segments and the cells. In the irrigation process, a canal segment may irrigate more than one surface cell because of the existence of lower-level canals. The same applies to the surface–ditch connection in the surface drainage process. The canal seepage occurs where it is, and so does the groundwater drainage, which means the connection between groundwater cells and canal/ditch segments only depends on their relative location.

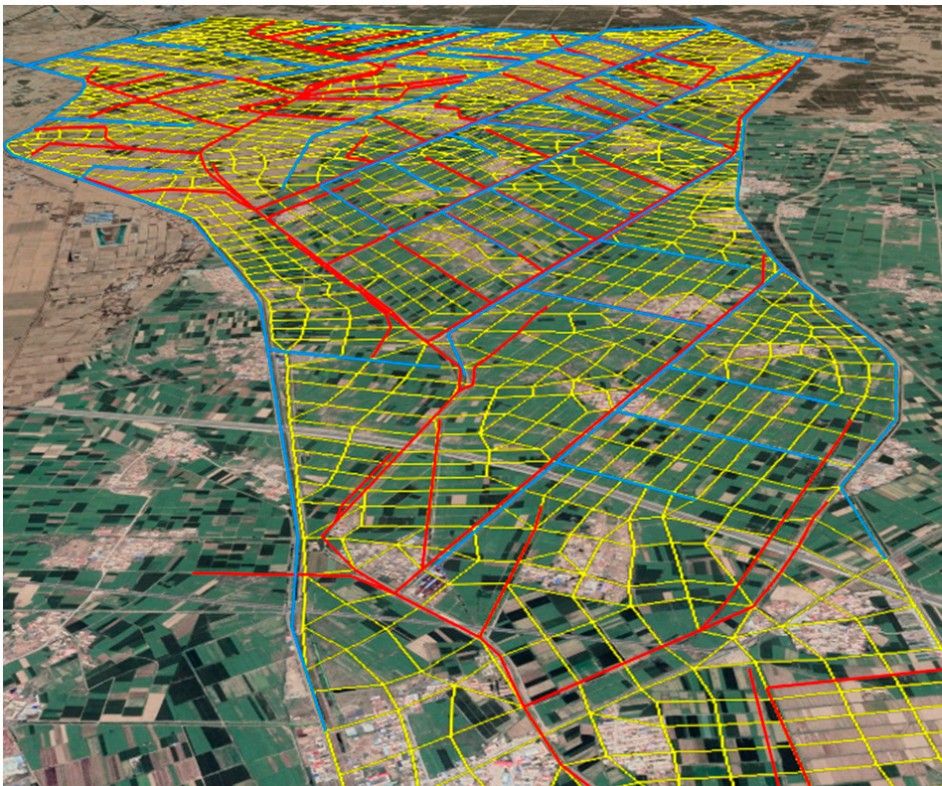

**Figure 4.** An example of spatial discretization. The red lines stand for canals, the blue lines stand for ditches, and the yellow grids stand for cells of the surface field, soil water, and groundwater.

The spatial connection information determines where to transfer the water in different entities, while the temporal discretization determines when to transfer. The governing equations are solved separately, meaning each of them has a date-time tracker, and a global clock is responsible for the synchronization. The groundwater moves slowly which gives it a relatively long time step, while in soil water, the time step is dynamically adjusted, usually smaller than that of the groundwater, according to the change rate of soil water condition. The time step of groundwater is taken as the communicating interval to avoid frequent exchange between soil water and groundwater, which may slow down the solving process. The soil water may solve multiple steps to catch up with one step of groundwater. The time step of the surface flow is even shorter due to its rapid change of flow status. The exchanging water amount will be accumulated during the communicating interval and transferred in the synchronizing process.

Considering that the soil water units are solved independently, a multi-thread technique is introduced to parallel the soil water movement process. The canal network and ditch network can also be solved simultaneously. Figure 5 shows a schematic structure of the model constructed in this study: the Irrigation Districts Model (IDM). There are three layers in the model. The basic layer provides basic services for the upper layers, including common components for programming-related tools such as the multi-thread support and file input/output utilities, and shared components for module-related tools such as the matrix solver and date–time controller. The core layer is the central part of the model, containing all the modules for water flow simulation and auxiliary components. The arrowed dense line means water exchange. The manage layer is in charge of the model coupling, data management, solving process control, output control, etc.

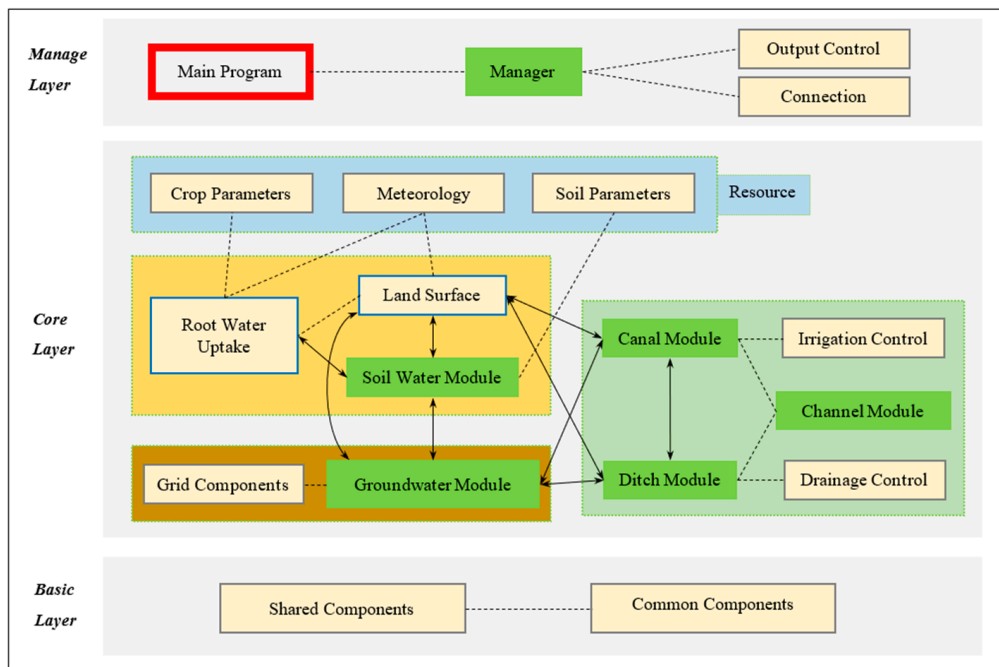

**Figure 5.** A schematic structure of Irrigation Districts Model (IDM).

*2.5. Model Inputs and Outputs*

The model inputs are mainly in correspondence with the core layer components and modules in Figure 5. The polygon grid information, including the coordinates of the polygon vertices, is required by the grid components to generate necessary parameters such as the cell area and the interpolating factors for ghost nodes. The land surface elevation for each cell will be used in both the groundwater module and soil water module. The land use in each cell is necessary for the unit division of the soil water module. The canal network and ditch network are represented by lists of from-A-to-B records that connect the segments one by one, and each segment contains its geometry information. The connection between different modules (e.g., canal segments and surface cells) is also written in the form of from-A-to-B pairs. Besides the spatial information, hydraulic parameters differ in these modules. The saturated hydraulic conductivity ($Ksg$) and the specific yield/storage ($\mu$) are for the groundwater module. The VG model parameters of each soil layer, including the residual water content ($\theta r$), the saturated water content ($\theta s$), the shape parameter ($\alpha$), the soil texture parameter ($n$), and the saturated hydraulic conductivity ($Kss$), are for the soil water module. Manning's roughness coefficient ($Nm$) is for the canal and ditch module. Other auxiliary components require more data to aid the module simulation. The meteorological information includes the daily minimal and maximum temperature, relative humidity, wind speed at a 2 m height, sunshine hour, precipitation, and pan evaporation. The parameters for each crop include the crop coefficient ($K_C$), the leaf area index ($LAI$), the root depth ($RD$), and the water stress parameters ($p$, $h_{50}$), which may vary with time to represent the crop growth. The irrigation information is given by round, and each irrigation round requires the round beginning and ending time, crops to be irrigated, the total irrigation water amount, and the seepage and discard ratio of canal chains. The drainage parameters mainly include the maximum allowed ponding depth for surface drainage and averaged lower-level ditches spacing for groundwater drainage.

The model outputs include the cumulative water exchange of processes illustrated in Figure 1, the groundwater level, the soil water content at specified layers, and the canal and ditch flow (channel water level and flow rate).

### 3. Test Case

To validate the model, experimental data from the Shahaoqu Sub-Irrigation Area (SHQ) were utilized for calibration and verification. SHQ is located in Jiefangzha (JFZ), a subarea of Hetao Irrigation District (HID) in Inner Mongolia of China, shown in Figure 6, with an area of 53.3 km², 66.5% of which is irrigated land, and the soil type is mainly silty loam. It is a typical arid region with approximately 150 mm of annual precipitation and 2000 mm of annual evaporation. The main crops here are sunflower (grown from late May to early October), maize (grown from late April to early October), and wheat (grown from early April to mid-July). There are five irrigation rounds during the crop growth period (from April to October) in a year, and the average water table depth is about 1.8 m during this period.

The experiment was conducted during 2018 and 2019. The water table depth was measured at 21 sites three times a month. The soil water content was measured at 13 sites with 5 layers in depth (20 cm per layer) once or twice a month. The 2018 data were used to calibrate the model parameters, and the 2019 data were used to verify the calibrated results. The simulation period is from 1st April to 1st October in both years.

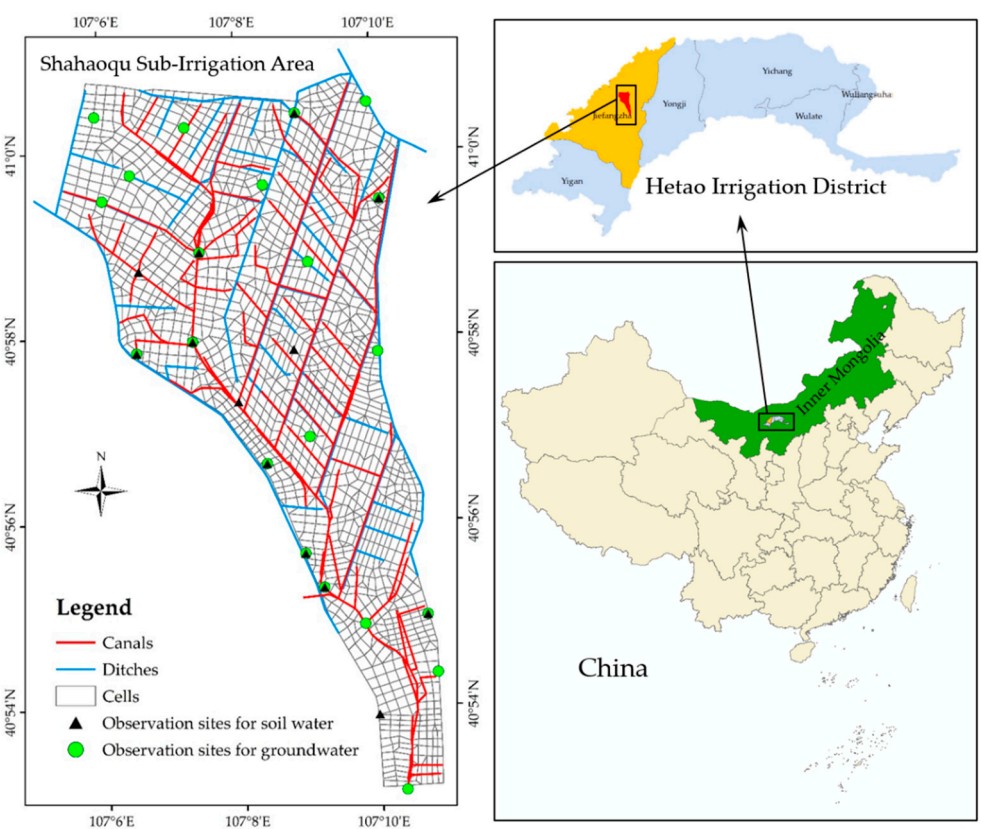

**Figure 6.** Location of Shahaoqu Sub-Irrigation Area and the distribution of canals, ditches, cells, and observation sites for soil water and groundwater.

#### 3.1. Input Data

The land surface elevation is important for spatial discretization. The 30 m DEM data from USGS (https://earthexplorer.usgs.gov, accessed on 29 August 2018) were adopted, after being calibrated with scattered points elevation measured by an RTK, to determine the averaged elevation in each cell. The surface elevation of SHQ is high in the south and low in the north, with a slope of around 0.02%. The irrigation water flows from south to north, and the drainage water flows out of the region through ditches at region boundaries. The distribution of canals, ditches, and cells in SHQ is shown in Figure 6. The land use data were processed from Gaofen-1 satellite data, and six types of land use are proposed: four

kinds of crops (sunflower, maize, wheat, and melon for melons and vegetables), wasteland, and others (houses, roads, etc.). Sunflower is the dominant crop in SHQ, accounting for nearly 70% of the area of all crops in 2018 and 90% in 2019. There are 1781 cells for groundwater and 3403 (2018) and 2833 (2019) units for soil water in these cells due to the different land use. The grid step for soil water is controlled at 5 cm, which generates an average of 100 vertical grids for each soil water unit. There are 71 canal chains and 42 ditch chains in total, which are divided into 650 canal segments and 496 ditch segments, respectively.

The calibrated hydraulic parameters are listed in Table 1. The original soil water parameters are based on Ren's work [46].

**Table 1.** Calibrated hydraulic parameters.

| Item | | Value | | | |
|---|---|---|---|---|---|
| Groundwater | $Ksg$ [m·d$^{-1}$] | 14.4 | | | |
| | $\mu$ [-] | 0.03 | | | |
| Soil water | Layer depth [cm] | 0~40 | 40~170 | 170~250 | 250~300 |
| | $\theta r$ [-] | 0.050 | 0.043 | 0.073 | 0.043 |
| | $\theta s$ [-] | 0.413 | 0.460 | 0.500 | 0.450 |
| | $\alpha$ [cm$^{-1}$] | 0.010 | 0.012 | 0.007 | 0.012 |
| | $n$ [-] | 1.567 | 1.467 | 1.200 | 1.700 |
| | $Kss$ [cm·d$^{-1}$] | 10.333 | 17.467 | 6.000 | 28.100 |
| Canal flow | $Nm$ [m$^{-1/3}$·s] | 0.022 | | | |
| Ditch flow | $Nm$ [m$^{-1/3}$·s] | 0.029 | | | |

The meteorological data are from the Linhe National Weather Station, 30 km away from SHQ. The cumulative precipitation in the simulation period is 157 mm in 2018 and 78 mm in 2019. The crop parameters are mainly based on relevant research in HID such as Hao's work [47]. The irrigation information is listed in Table 2.

**Table 2.** Irrigation information.

| Year | Round | Time | Water Amount [10$^4$ m$^3$] | Crops to Irrigate |
|---|---|---|---|---|
| 2018 | 1 | 4/23~5/14 | 577.4 | wheat, sunflower, melon |
| | 2 | 5/14~5/25 | 365.6 | wheat, sunflower |
| | 3 | 6/13~6/26 | 368.3 | wheat, maize, sunflower |
| | 4 | 7/3~7/19 | 398.9 | maize, sunflower |
| | 5 | 7/26~8/4 | 173.6 | maize, sunflower |
| 2019 | 1 | 4/28~5/16 | 530.4 | wheat, sunflower, melon |
| | 2 | 5/16~5/30 | 500.7 | wheat, sunflower |
| | 3 | 6/14~6/23 | 194.6 | wheat, maize |
| | 4 | 7/6~7/20 | 424.8 | maize, sunflower |
| | 5 | 7/26~8/9 | 259.1 | maize, sunflower |

A water level gauge was put at the end of the ditches to monitor the water level of drainage, which will be used as the boundary values for the ditch network. The measured groundwater level gradient at the region boundary is used for the boundary condition of groundwater.

*3.2. Simulation Results*

The simulation begins on 1st April and ends on 1st October in both 2018 and 2019, which covers the crop growth period. There are five irrigation rounds during this period, starting in late April and ending in early August. A 15-day pre-simulation is applied to reduce the impact of limited knowledge of initial conditions on the simulating results. Table 3 shows the overall water balance in the study area of the two years. The water balance error of the whole region, including the error in each module and the error of

module coupling, is controlled under 3% in both years. The water balance in the soil layer of a 1 m depth from the surface is shown in Table 4. The soil water storage increases $2.1 \times 10^4$ m$^3$ in 2018 but decreases $169.6 \times 10^4$ m$^3$ in 2019. The net recharge from a 1 m depth below is $196.5 \times 10^4$ m$^3$ and $309.5 \times 10^4$ m$^3$ in 2018 and 2019, respectively.

**Table 3.** The overall water balance (unit: $10^4$ m$^3$).

| Item | | 2018 | 2019 |
|---|---|---|---|
| Precipitation | | 827.7 | 408.8 |
| | Inlet | 1820.9 | 1834.3 |
| | To fields | 1197.3 | 1210.7 |
| Irrigation | Seepage | 513.3 | 513.5 |
| | Evaporated | 18.7 | 11.4 |
| | Discarded | 93.8 | 100.9 |
| Surface evaporation | | 883.8 | 856.8 |
| Crop transpiration | | 1335.0 | 1241.8 |
| Groundwater drainage | | 218.8 | 222.4 |
| Groundwater boundary flow [1] | | −119.7 | −119.9 |
| Water storage change [2] | | −71.1 | −337.1 |
| Balance error [3] | | 52.1 (2.9%) | 29.1 (1.6%) |

[1] flow through lateral boundaries, negative for flowing out of the region. [2] storage for both soil water and groundwater, negative for storage reduction. [3] percentage to irrigation water amount form the inlet.

**Table 4.** Water balance of the 1 m-depth soil layer (unit: $10^4$ m$^3$).

| Item | 2018 | 2019 |
|---|---|---|
| Precipitation | 827.7 | 408.8 |
| Irrigation | 1197.3 | 1210.7 |
| Evapotranspiration | 2218.8 | 2098.6 |
| Net recharge from 1 m depth below | 196.5 | 309.5 |
| Water storage change [1] | 2.7 | −169.6 |

[1] storage for the 1m-depth soil layer, negative for storage reduction.

Four indexes are adopted to estimate the simulated results. The Nash–Sutcliffe model efficiency coefficient is

$$NSE = 1 - \frac{\sum_{i=1}^{n}(S_i - O_i)^2}{\sum_{i=1}^{n}(O_i - O_{ave})^2} \tag{14}$$

where $n$ is the number of the values, $S_i$ is the $i$-th simulated value, $O_i$ is the $i$-th observed value, and $O_{ave}$ is the average of observed data. The relative error is

$$RE = \frac{\sum_i(S_i - O_i)}{\sum_i O_i} \tag{15}$$

The mean residual ratio is

$$MRR = \frac{\frac{1}{n}\sum_{i=1}^{n}|S_i - O_i|}{(O_{max} - O_{min})} \tag{16}$$

where $O_{max}$ is the maximum observed value, and $O_{min}$ is the minimum observed data. The root mean square ratio is

$$RMSR = \frac{\sqrt{\frac{1}{n}\sum_{i=1}^{n}(S_i - O_i)^2}}{(O_{max} - O_{min})} \tag{17}$$

The NSE is used to assess the predictive skill of the model, with a range of $(-\infty, 1)$. The closer it is to 1, the better the results it suggests. The RE, MRR, and RMSR are used to describe the error condition. The closer it is to 0, the smaller the error.

The simulated water table depth at 21 sites is compared with the observed data in Figure 7. The simulated soil water content of 5 layers at 13 sites was compared with the observed data, as shown in Figure 8. The result shows a good simulation of the water movement in irrigation districts with IDM.

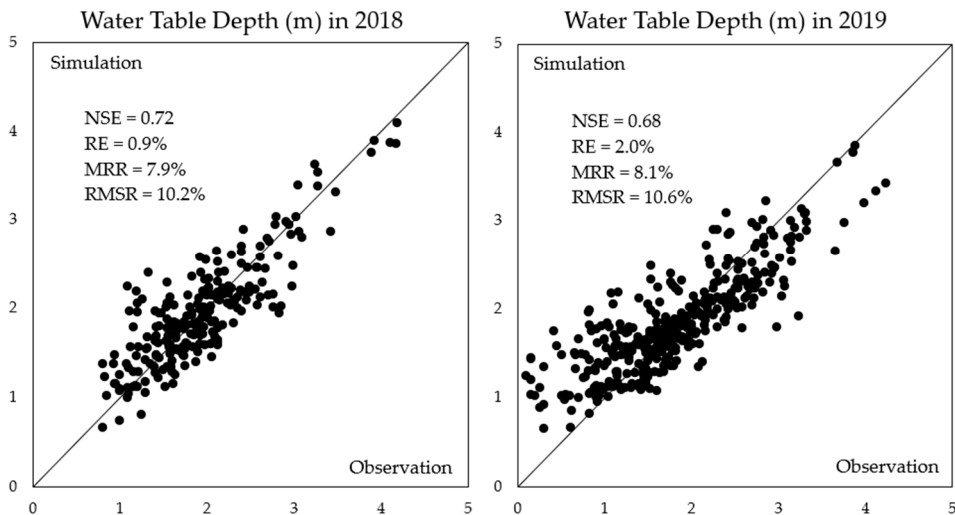

**Figure 7.** Comparison between simulated and observed water table depth.

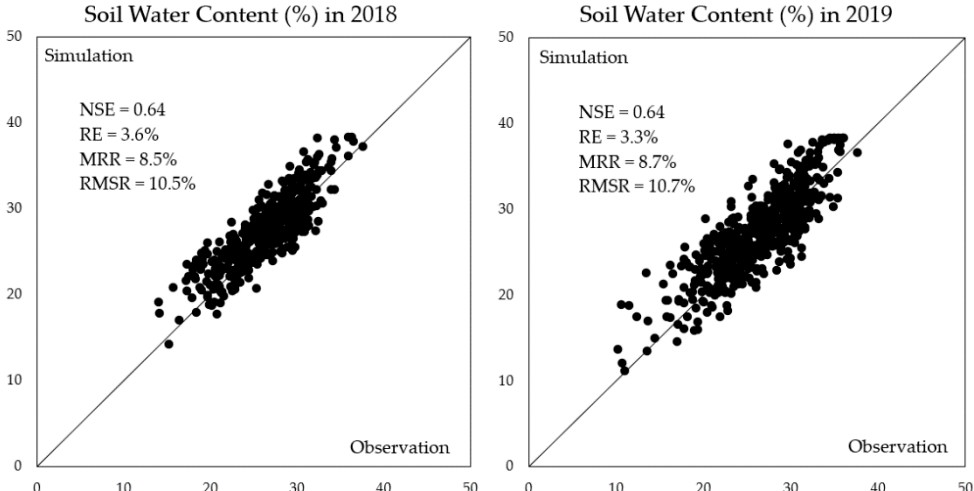

**Figure 8.** Comparison between simulated and observed soil water content.

Figure 9 shows the simulated groundwater level process at one observation site and the corresponding observed data, along with the precipitation and irrigation process. The simulation has good correspondence with the observation. The groundwater level rises after the irrigation and reaches its peak after the second irrigation round in late May. It maintains a relatively high level during the irrigation period, which is from early May to early August and then continues to drop after the last irrigation round. The rainfall can also cause the groundwater level rise, especially the intense one, but with less impact compared to the irrigation.

The simulated soil water content process at each layer of an observation site is compared with the observed data in Figure 10, which shows good correspondence. The soil water content increases gradually from the top layer down. The upper layers experience greater fluctuation than the lower layers.

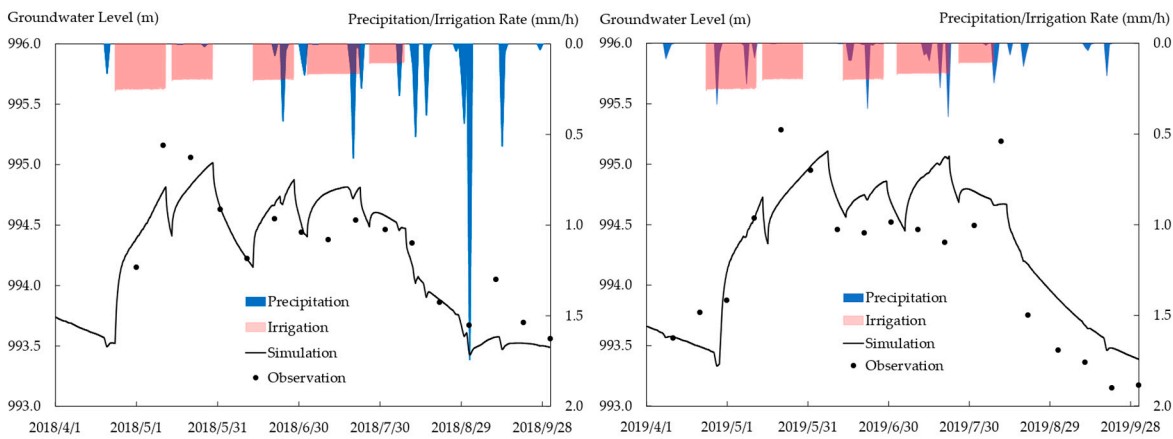

**Figure 9.** Process of precipitation and irrigation, and the simulated and observed groundwater level at an observation site.

**Figure 10.** Process of simulated and observed soil water content at 5 layers of an observation site.

The first two irrigation rounds are continuous and account for half of the total irrigation amount in 2018 (see Table 2). The groundwater level change before and after the first two irrigation rounds in 2018, which is from 23/4/2018 to 25/5/2018, is shown in Figure 11. The groundwater level climbs up along the canals after the irrigation, which demonstrates the impact of canal distribution on the regional water movement.

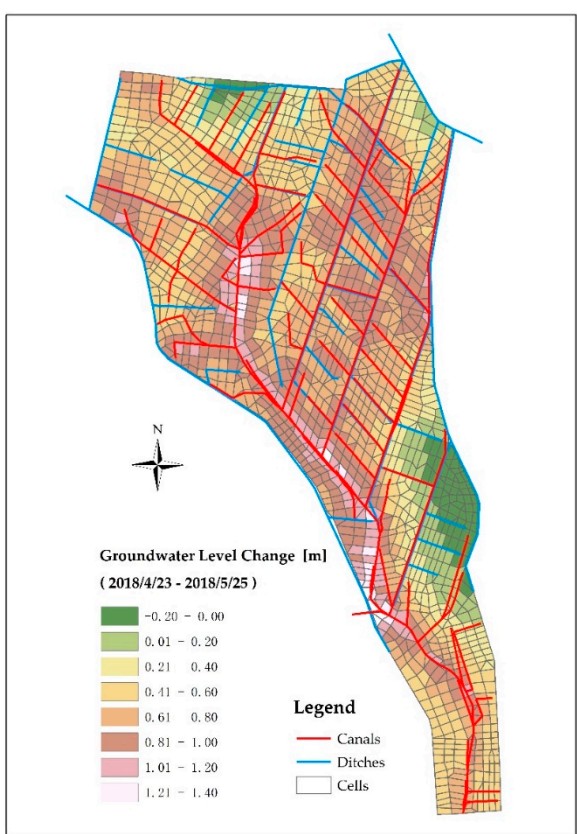

**Figure 11.** Groundwater level change before and after the first two irrigation rounds in 2018.

To test the effect of multi-thread computing, a personal computer with a 10-core CPU is used to run the same scenario with different numbers of threads. The computing time is divided by a base value, which is the total computing time with one thread (78 min in the test), and the results are listed in Table 5.

**Table 5.** Relative calculating time consumed with different numbers of threads (all the computing times are relative to the total computing time with one thread).

| Number of Threads | 1 | 2 | 4 | 8 |
|---|---|---|---|---|
| Total time consumed | 1 | 0.75 | 0.50 | 0.35 |
| Time for canals | 0.01 | 0.01 | 0.01 | 0.01 |
| Time for ditches | 0.01 | 0.01 | 0.01 | 0.01 |
| Time for soil water | 0.81 | 0.56 | 0.33 | 0.19 |
| Time for groundwater | 0.08 | 0.08 | 0.08 | 0.08 |
| Time for output | 0.10 | 0.08 | 0.06 | 0.06 |

## 4. Discussion

The water storage change of the whole region, including soil water and groundwater, between 2018 and 2019 is substantial. The main reason is that the precipitation amount in 2018 is about twice that of 2019. There are several heavy rains in August 2018, most of which concentrate after the last irrigation round when the groundwater level drops fast (see Figure 9). The rainfall mainly recharges the unsaturated soil part at a low level

of groundwater, which explains why the groundwater drainage is not affected by the precipitation and the water storage in a 1-m-depth soil layer increases slightly in 2018 while decreasing substantially in 2019.

The groundwater level peak appears after the second irrigation round (see Figure 9), because the first two irrigation rounds are continuous with large amounts of water, and the soil water consumption by crops remains at a low level during the initial period of crop growth. The groundwater level climbs up rapidly and then stays at a relatively high level dynamically due to the balance of irrigation and crop absorption. After the last irrigation round, the groundwater level drops down gradually. During the first two irrigation rounds, the observed water table depth becomes shallower (less than 1 m), but the simulated result remains deep, which causes the deviation in 2019 of Figure 7. The reason is the generalization of the irrigation process. The irrigation flow rate is limited by the canal size, and the whole region is irrigated piece by piece, not all at once. The canal flow is manually controlled and the regulation is usually out of record. Because of the lack of detailed irrigation regulation, it can only be generalized that all the fields with crops to be irrigated will receive the irrigation simultaneously as soon as water flows to the neighboring canal segments. The result of this generalization is that the centralized short-time irrigation at one field is prolonged to the period of an irrigation round with the same irrigation water demand. A longer irrigation time means a smaller irrigation rate, so the water recharge from soil water to groundwater will be reduced, and the groundwater level will rise slowly and fail to reach the peak of the observation. The same applies to precipitation. The meteorological information is given by the day, while a heavy rain only lasts for hours. The average rainfall will smooth out the peak effect, as shown in Figure 9, with the heaviest rain on 1st September 2018. The groundwater level rises about 0.5 m after the rain based on observations, but there is only a 0.1 m rise in the simulated results. More detailed information is required for a better result.

The soil water content is simulated in independent soil columns of the region. It is highly affected by the surface boundary, as we can see in Figure 10. The soil water content of the surface layer increases rapidly when there is irrigation or precipitation, and then decreases gradually due to the crop root absorption, the surface evaporation, and the water exchange with the bottom layer. The balance between water supply and water consumption causes the variation of soil water content. The same applies to bottom layers except that the water supply is from neighboring layers and there is no evaporation. The change rate of soil water content at bottom layers is gentler than that at top layers, because the water movement in soil is quite slow and the rapid change of the surface boundary will be diminished as going deeper in soil. At the layer of 80–100 cm, the soil water content is rather stable. It is important to describe the soil water movement hydraulically so that the crop root water absorption and the water exchange among surface water, soil water, and groundwater can be characterized in detail. Crops are vital for irrigation districts, and the soil water condition is vital for crop growth.

The canal distribution has a great impact on regional water movement. The groundwater level climbs up differently after irrigation due to the different land use, as shown in Figure 11. Furthermore, the impact of canal seepage on groundwater level cannot be ignored, as we can see the white spots along the main canal in Figure 11, which shows the greatest groundwater level change. Without these canals, the detailed water distribution in irrigation districts is difficult to describe.

Multi-thread computing is efficient, especially for the soil water simulation, which occupies most of the computing time. However, the computing speed is not linear to the number of threads because of the extra overhead required by thread synchronization. The Dupuit assumption adopted in groundwater movement has its limitations. It is a proper simplification at plain irrigation districts where the groundwater level is flat as the terrain. However, in mountain areas with a significant vertical flow of groundwater, the theoretical error might be introduced. The soil water module and groundwater module are

loosely coupled at the current version of IDM, which may introduce a small balance error. However, the overall balance error is acceptable according to the test cases.

The water movement in irrigation districts is much more complicated than in the natural region, because we focus more on the unsaturated soil water part that affects the crop directly, and have more artificial regulations to control the water condition in irrigation districts. The current hydrological models fail to take all the processes in irrigation districts into account, especially the irrigation process, which proved to have a great impact on regional water movement. The main advantage of the model proposed in this study is that it comprehensively considers water movement in irrigation districts and afford improved simulation. The disadvantage, however, is obvious: detailed consideration requires detailed data support. It is important to balance both the functionality and data simplicity of the model. Moreover, the water movement is closely related to the salt transport in arid/semi-arid irrigation districts, which affects the growth environment of crops. The salt movement based on the water movement will be the focus of the following work.

## 5. Conclusions

A novel physically based distributed model (IDM) specialized for the water movement in irrigation districts is proposed in this study. The model fully considers all the important processes in irrigation districts, including the irrigation process, soil water movement, groundwater movement, the drainage process, and the water interactions in these processes. The water flow in irrigation canals and drainage ditches is governed by the 1D diffusive wave equation. The soil water movement is governed by the 1D Richards equation and integrated into the regional scale. The groundwater movement is governed by the 2D groundwater equation, which is based on the Dupuit assumption. The irrigation district is divided into polygons (cells) according to the land use and distribution of canals and ditches, and the soil water units are connected with these groundwater cells to exchange water. The irrigation is carried out by round, and only the field with the proper crop will be irrigated. The model was validated with experimental data. The 2018 data were used to calibrate the model parameters, and the 2019 data were used to verify the calibrated parameters. The overall water balance error was under 3%, which is acceptable. The simulated water table depth and the soil water content were compared with the observed data, which showed good correspondence. Multi-thread computing was also tested, and the computing speedup was obvious. The results suggest that the model is capable of simulating the water movement in an irrigation district. It also provides a functional tool for water movement research in irrigation districts.

**Author Contributions:** Conceptualization, H.C. and B.M.; methodology, H.C. and B.M.; software, B.M.; validation, H.C. and B.M.; investigation, B.M. and Y.J.; resources, J.J., X.C., X.F., R.C., and M.W.; writing—original draft preparation, B.M.; writing—review and editing, S.W. and H.C. All authors have read and agreed to the published version of the manuscript.

**Funding:** This work was funded by the National Key Research and Development Program of China (2019YFC0409203, 2016YFC0501301, 2017YFC0403302, 2017YFC0403205), National Natural Science Foundation of China (51779273), and Basic Research Fund of China Institute of Water Resources and Hydropower Research (GG0145B502017).

**Institutional Review Board Statement:** Not applicable.

**Informed Consent Statement:** Not applicable.

**Data Availability Statement:** The data presented in this study are available on request from the corresponding author. The data are not publicly available due to research continuation.

**Acknowledgments:** The experiment was conducted in Shahaoqu Station with much help from the staff there, and their work is appreciated. Shaohui Zhang from China Institute of Water Resources and Hydropower Research, Xu Xu from China Agricultural University, and Yuanyuan Zha from Wuhan University have shared their knowledge about model construction, which was of great assistance.

Yuanyuan Zha helped to review the original draft and provided us with valuable advice. We would like to express our gratitude to all who supported us in this study.

**Conflicts of Interest:** The authors declare no conflict of interest.

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
