# Peer review of "A Novel Physically Based Distributed Model for Irrigation Districts’ Water Movement"

_water, doi:10.3390/w13050692_

Round 1
Reviewer 1 Report
1) What equipment was used to measure the position of grundwater table nad soil moisture values at different soil depth ?
2) What calibration equation was used in this device ? (from the producer, from the literature or own calibration formula)
3) There is no description of the type of soil on the site and some physical and hydraulic properties
4) Were the own measurements of soil moisture carried out , e.g. using for example gravimetric method in order to compare the obtained results from indirect method ?
5) On the basis of such short calibration period of 1 year (avarage value of rainfall - 2018) and 1 year of validation (dry year - 2019). Can it be concluded that the model can be used and valid for normal and dry years ?. Such models are constructed during at least 3-4 years for similar meteorological conditions. Some paramaters were performed (calibrated) for normal year and than used for dry year ?
6) Soil moisture content once or twice a month in my opinion are too infrequent, they should be measured continuously or at least one or twice a week.
7) The differences between the calculated and measured water table position are too great (fig.7) , especially in the shallower position range of water table. This is especially visible for dry year 2019 (fig. 9). Generally they should be much more similar (observed and calculated by model).
8) In fig. 8 the differences between the calculated and measured soil moisture content values are approx. 8-10%, while in fig. 10 these differences are suprisingly low ?, how it can be explained.
In my opinion in such type of models the values of calculated and simulated water table position should be quite close, but the values of calculated and measured values of moisture content show quite big differences.
Reviewer 2 Report
I appreciate the Editor to give me a chance to review an interesting and valuable paper. I found some merits in the both methodology and results. In my opinion, this paper has a good potential to be published in the journal. However, I have also some concerns on the different parts of the manuscript. If the author(s) address carefully to the comments, I’ll recommend publication of the manuscript in the journal:
- Add/Replace the name of the study area to the Keywords.
- In the last paragraph of the Introduction, the authors should clearly mention the weakness point of former works (identification of the gaps) and describe the novelties of the current investigation to justify us the paper deserves to be published in this journal.
- Cite these recent useful papers on groundwater modeling to improve the literature and to show the importance of your work:
Identifying Optimal Locations for Artificial Groundwater Recharge by Rainfall in the Kingdom of Bahrain
Groundwater Resources Management of the Shallow Groundwater Aquifer in the Desert Fringes of El Beheira Governorate, Egypt
Mapping of Groundwater Recharge Potential Zones and Identification of Suitable Site-Specific Recharge Mechanisms in a Tropical River Basin
- Focus on the main reasons of the variations of the simulated and observed soil water content at 5 layers of an observation site.
- What are the strategies/recommendations to reduce uncertainties in this study?
- How can extend the results in other regions with similar/different groundwater conditions?
- The quality of the language needs to improve by a native English speaker for grammatically style and word use.
